# Rad52's DNA annealing activity drives template switching associated with restarted DNA replication

Anastasiya Kishkevich [1], Sanjeeta Tamang[1], Michael O. Nguyen[1], Judith Oehler [1], Elena Bulmaga [1], Christos Andreadis[1], Carl A. Morrow [1], Manisha Jalan[1], Fekret Osman[1] & Matthew C. Whitby [1] ✉

It is thought that many of the simple and complex genomic rearrangements associated with congenital diseases and cancers stem from mistakes made during the restart of collapsed replication forks by recombination enzymes. It is hypothesised that this recombination-mediated restart process transitions from a relatively accurate initiation phase to a less accurate elongation phase characterised by extensive template switching between homologous, homeologous and microhomologous DNA sequences. Using an experimental system in fission yeast, where fork collapse is triggered by a site-specific replication barrier, we show that ectopic recombination, associated with the initiation of recombination-dependent replication (RDR), is driven mainly by the Rad51 recombinase, whereas template switching, during the elongation phase of RDR, relies more on DNA annealing by Rad52. This finding provides both evidence and a mechanistic basis for the transition hypothesis.

The course to complete genome duplication is strewn with obstacles, including various DNA lesions and DNA binding proteins, which threaten the successful completion of DNA replication by waylaying replications forks and, occasionally, causing their collapse[1]. Fork collapse involves the disassembly or remodelling of the replisome rendering it unable to continue DNA synthesis[2]. The consequent loss of replicative capacity can lead to mitotic catastrophe through the attempted segregation of incompletely replicated chromosomes. To avoid this, cells deploy homologous recombination enzymes to repair collapsed forks and restart DNA replication in a process termed recombination-dependent replication (RDR) or break-induced replication (BIR) if initiated from a DNA double strand break (DSB)[3,4].

Mechanistic understanding of RDR in eukaryotes has come mainly from studies of BIR in the budding yeast *Saccharomyces cerevisiae*, which have identified two distinct pathways that share a requirement for the Rad52 recombination protein, but differ in their reliance on a number of other recombination proteins, most notably Rad51[5–8]. In budding yeast, Rad51-dependent BIR is the predominant pathway exhibiting a specific requirement for Rad51, the Swi2/Snf2 family protein Rad54, and the Rad51 paralogues Rad55 and Rad57. It proceeds

via nucleolytic resection of the DSB, which generates a 3′-OH ended single-stranded (ss) DNA tail that is initially bound by Replication Protein A (RPA). Rad52 then mediates the replacement of RPA by Rad51, which forms a nucleoprotein filament with the DNA[5]. With the help of Rad54, this filament locates and invades a homologous DNA template to form a displacement (D) loop at which new DNA synthesis is primed[9,10]. DNA replication then proceeds via leading strand synthesis within the context of a migrating D-loop or bubble generating an extensive region of nascent ssDNA that is subsequently used as the template for lagging strand synthesis[11,12]. Therefore, unlike canonical DNA replication, BIR is conservative rather than semi-conservative.

In contrast to Rad51-dependent BIR, Rad51-independent BIR is a minor and inefficient pathway in budding yeast and, consequently, has received relatively little attention and remains poorly understood[4,13]. Nevertheless, certain features of the Rad51-independent pathway have been documented, including: a dependence on Rad52, Rad59, Rdh54 and the Mre11-Rad50-Xrs2 complex; its suppression by Rad51; and a reduced requirement for homology between broken DNA and template relative to the Rad51-dependent pathway[7,8,14]. Moreover, although the mechanism for the initiation of Rad51-independent BIR

[1]Department of Biochemistry, University of Oxford, South Parks Road, Oxford OX1 3QU, UK. ✉e-mail: matthew.whitby@bioch.ox.ac.uk

remains unknown, it is thought to rely on Rad52's DNA annealing activity operating between the resected broken DNA and another region of homologous or homeologous ssDNA exposed at a replication fork, transcription complex or secondary DNA structure[15]. Whilst Rad51-independent BIR may be a minor pathway in budding yeast, recent evidence suggests that it plays a more prominent role in human cells. For example, a form of Rad51-independent BIR termed MiDAS (mitotic DNA synthesis) helps ensure that difficult to replicate genomic regions, such as common fragile sites, are fully replicated before anaphase completes[16,17]. And cancerous cells experiencing high levels of replication stress, due to oncogene overexpression, appear to rely on Rad51-independent BIR for their survival[18]. These findings highlight the importance of understanding how Rad51-independent RDR works and coordinates with the Rad51-dependent pathway.

Studies in budding yeast have shown that Rad51-dependent BIR is associated with much higher rates of mutations and genome rearrangements than canonical DNA replication[4,13]. This has been attributed to: (1) its conservative mode of DNA synthesis that renders mismatch repair inefficient[12,19]; (2) the uncoupling of leading and lagging strand synthesis resulting in tracts of ssDNA whose replication becomes highly mutagenic when damaged[20]; and (3) template switching due to frequent dissociation of the migrating D-loop coupled with re-invasion of the ejected DNA strand at ectopic homologous or homeologous sites[21–23]. The inherent mutagenicity of BIR is thought to contribute to the development of many genetic diseases and, in particular, template switching is likely responsible for some of the simple and complex genome rearrangements associated with various cancers and other genomic disorders[24,25]. Such rearrangements exhibit varying amounts of sequence homology at their breakpoint junctions ranging from short stretches of only 2–30 bp (classified as microhomology) to regions >100 bp. The frequent occurrence of only microhomology at breakpoint junctions is thought to be incompatible with a Rad51-dependent process, as efficient strand invasion by Rad51 in vivo typically requires about 70 bp of homology[26]. This prompted Hastings and colleagues to propose a variant BIR pathway termed microhomology-mediated BIR (MMBIR), which may be synonymous with Rad51-independent BIR and occur under conditions, such as hypoxia, where Rad51 is repressed[27]. Based on analysis of the complex genomic rearrangements found in patients with MECP2 duplication syndrome and Pelizaeus-Merzbacher disease and the findings of genetic studies in budding yeast, it became apparent that, rather than being completely separate processes, BIR and MMBIR may be intrinsically linked, with the former transitioning into the latter following disruption of repair DNA synthesis[22,28,29]. However, evidence that Rad51-dependent BIR/RDR can transition into a Rad51-independent process that relies on Rad52 to drive template switching is lacking.

In this study we use the fission yeast model system to investigate the contribution of Rad51-dependent and -independent pathways to replication restart and template switching triggered by replication fork stalling and collapse at a site-specific protein-DNA barrier. We show that these pathways can operate alongside each other to drive restart and template switching. We also provide evidence that template switching becomes more reliant on Rad52's DNA annealing activity as RDR progresses from the site of fork collapse consistent with the model that BIR/RDR transitions from a Rad51-dependent to Rad51-independent process.

## Results

### Measuring ectopic recombination and template switching associated with RDR

We have previously developed assays in the fission yeast *Schizosaccharomyces pombe* for measuring ectopic recombination linked to the initiation of RDR (Fig. 1a) and template switching associated with its elongation phase (Fig. 1b)[30,31]. These assays utilise the polar replication fork barrier (RFB) *RTS1*, which strongly blocks replication forks

and triggers their collapse without inducing a DSB[31,32]. It is thought that the collapsed fork reverses creating a free duplex DNA end, which is processed into a 3′-OH ended ssDNA tail that is ultimately bound by Rad51[31,33,34]. Like in BIR, Rad51 then catalyses the invasion of a homologous duplex DNA, by the ssDNA that it is bound to, generating a D-loop at which new DNA synthesis is primed (Fig. 1a, b). This whole process, from fork collapse to the initiation of RDR, is estimated to take ~20 min[31,35,36]. In our experimental system, *RTS1* is inserted downstream of a cluster of strong early firing replication origins on chromosome 3, making its replication essentially unidirectional (Fig. 1c). As *RTS1* is a polar RFB, it only blocks replication forks at this site when inserted in what we term its Active Orientation (AO). In the opposite or Inactive Orientation (IO), replication forks pass *RTS1* relatively unhindered resulting in no induction of recombination[37]. Therefore, strains containing *RTS1*-IO are used to assess background spontaneous recombination. A genetic reporter consisting of a direct repeat of mutant *ade6* genes and intervening *his3⁺* gene flanking *RTS1* (0 kb reporter) is used to measure ectopic recombination associated with the initiation of RDR, whereas the same reporter positioned 12.4 kb downstream of *RTS1* (12.4 kb reporter) measures template switching associated with RDR as it progresses from its initiation site (Fig. 1c, d). In both cases the reporter registers Ade⁺ recombinants that arise by deletion and gene conversion events (Fig. 1d).

### Ectopic recombination associated with RDR initiation is driven mainly by Rad51-mediated strand invasion

To determine the contribution of the Rad51-dependent pathway to ectopic recombination associated with RDR initiation, we measured the frequency of Ade+ recombinants, using the 0 kb reporter, in strains deleted for Rad51 or one of its known auxiliary factors. Differences between spontaneous and *RTS1*-induced recombination were assessed by comparing strains with *RTS1*-IO (Fig. 2a) and *RTS1*-AO (Fig. 2b). Consistent with previous data, recombination is strongly induced by *RTS1*-AO with gene conversions and deletions increasing ~96-fold and ~25-fold ($p$ values ≤ 0.0001) respectively, compared to spontaneous levels[31,37]. Spontaneous gene conversions are largely abolished in a *rad51Δ* mutant and *RTS1*-AO induced gene conversions decrease by ~97-fold ($p$ value ≤ 0.0001) consistent with their dependence on Rad51-mediated strand invasion. In contrast, spontaneous deletions increase ~5-fold ($p$ value ≤ 0.0001) and the frequency of *RTS1*-AO-induced deletions remains similar to wild-type ($p$ value = 0.0802). Whilst the increase in spontaneous deletions and reduction in spontaneous and *RTS1*-AO induced gene conversions are consistent with published data, previous studies have reported a ~1.5-fold to ~2.5-fold reduction in *RTS1*-AO-induced deletions in a *rad51Δ* mutant compared to wild-type[37–39].

The increase in spontaneous deletions and most of the residual *RTS1*-AO-induced recombination in a *rad51Δ* mutant are dependent on Rad52 (Fig. 2a, b)[37]. To see if this dependence requires Rad52's DNA annealing activity, we mutated conserved arginine-45 to alanine, which disables Rad52's strand annealing activity without hampering its Rad51 mediator function[40–42]. The *rad52*-R45A mutation has no statistically significant effect on the frequency of spontaneous recombination or *RTS1*-AO-induced recombination (Fig. 2a, b), indicating that Rad52's DNA annealing activity is not required for driving ectopic recombination associated with RDR initiation. In contrast, analysis of a *rad51Δ rad52*-R45A double mutant shows that the increase in spontaneous deletions in a *rad51Δ* mutant, and residual *RTS1*-AO-induced deletions and gene conversions, depend on Rad52's DNA annealing activity (Fig. 2a, b).

Correct Rad51 nucleofilament formation and function is reliant on auxiliary proteins, accordingly, *rad54Δ*, *rad54*-K300A (encoding ATPase defective Rad54[43–45]; see Supplementary Fig. 1), *rad55Δ* and *rad57Δ* mutants all exhibit a similar recombination phenotype to a *rad51Δ* mutant, though with one notable exception: the frequency of

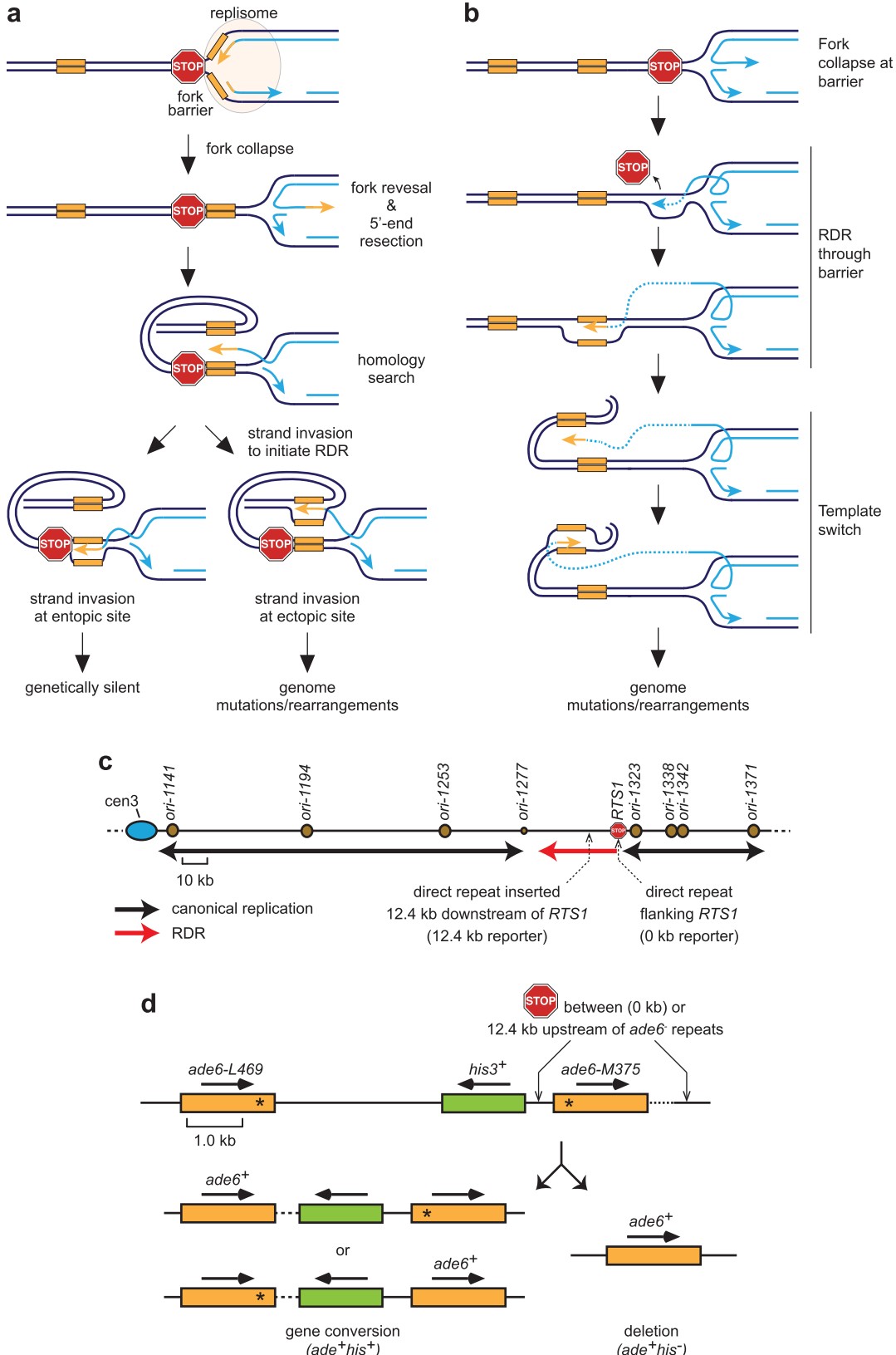

**Fig. 1 | Experimental system for measuring RDR-associated template switching.** **a** Model for ectopic recombination between repetitive DNA induced by fork collapse at a RFB. Parental DNA strands are in dark blue, nascent DNA strands are light blue, directly repeated DNA elements are in yellow and the RFB is indicated by the stop symbol. **b** Model for template switching between direct repeats associated with RDR. **c** Map of the chromosome 3 region where *RTS1* and 0 kb and 12.4 kb reporters are inserted. Replication origins are indicated by brown circles. Black and red arrows indicate the direction of canonical replication and RDR, respectively. **d** Schematic of the genetic reporter used to monitor ectopic recombination and template switching associated with RDR. Arrows indicate the direction of transcription of the *ade6* and *his3* genes, the Stop symbol indicates *RTS1*, and the asterisks indicate the position of point mutations in *ade6-L469* and *ade6-M375*.

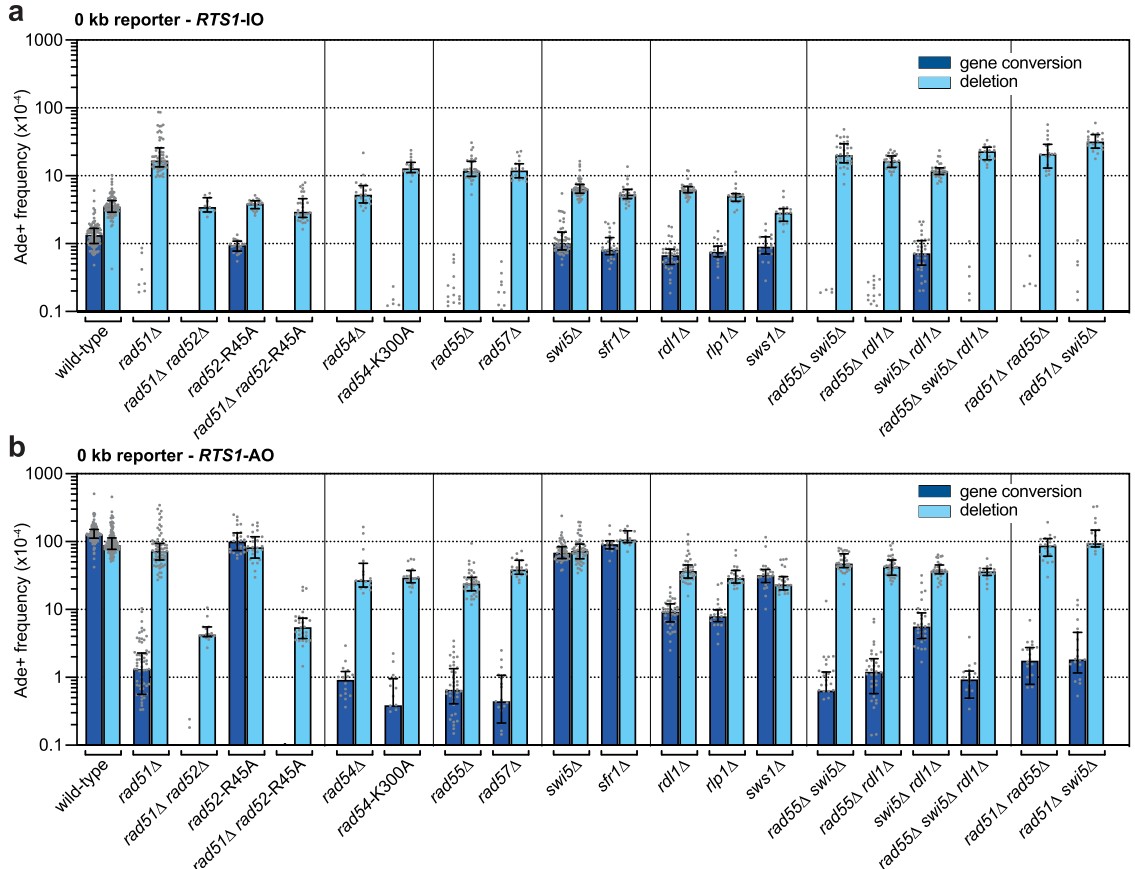

**Fig. 2 | Ectopic recombination associated with RDR initiation is driven mainly by Rad51 and its auxiliary factors. a** Frequency of spontaneous (*RTS1*-IO) Ade⁺ recombinants in wild-type and mutant strains carrying the 0 kb reporter. **b** Frequency of *RTS1*-AO-induced Ade⁺ recombinants in wild-type and mutant strains carrying the 0 kb reporter. **a,b** Data are presented as median values ± interquartile range with individual data points shown as grey dots. The data are also reported in Supplementary Table 1, which includes the strain numbers, the number of colonies tested for each strain (*n*) and *p* values. Further details of the statistical analysis are reported in Supplementary Data 1. Source data are provided as a Source Data file.

*RTS1*-AO-induced deletions is reduced by ~2–4-fold (*p* values = 0.0067–<0.0001) (Fig. 2a, b). Strains deleted for the Rad51 paralogues, Rdl1 and Rlp1, and the SWIM-domain containing protein Sws1 also exhibit similar fold reductions in *RTS1*-AO-induced deletions although gene conversions are reduced to a lesser extent (Fig. 2b). The equivalent proteins in budding yeast form the so-called Shu complex that interacts with Rad55 and is thought to function in replication-associated DNA repair alongside Rad51 and Rad52[46–48]. A *rad55Δ rdl1Δ* double mutant exhibits a similar phenotype for both spontaneous and *RTS1*-AO-induced recombination as a *rad55Δ* single mutant (Fig. 2b), which accords with Rdl1 and Rad55 functioning in the same pathway for promoting *RTS1*-AO-induced recombination. In contrast, mutation of the Swi5-Sfr1 complex, which reportedly acts in a parallel and separate pathway to Rad55-Rad57 in promoting Rad51 activity[49–52], has relatively little or no effect on spontaneous or *RTS1*-AO-induced recombination and is also not required for the residual recombination in either a *rad55Δ* or *rad55Δ rdl1Δ* mutant (Fig. 2a, b).

Even when compared to previous data[37–39], the fold reductions in *RTS1*-AO-induced deletions in *rad54Δ*, *rad54*-K300A, *rad55Δ*, *rad57Δ*, *rdl1Δ*, *rlp1Δ* and *sws1Δ* mutants are the same or greater than in a *rad51Δ* mutant. This observation, that a *rad51Δ* mutant tends to exhibit higher levels of *RTS1*-AO-induced deletions than its auxiliary factor mutants, suggests that the presence of Rad51 can partially suppress Rad52-mediated recombination as observed in budding yeast[15]. This notion is supported by the finding that a *rad51Δ rad55Δ* double mutant exhibits a similar frequency of *RTS1*-AO-induced recombination as a *rad51Δ* single mutant (*p* value ≥ 0.9999) (Fig. 2b). In addition, a *rad51*-R152A-R324A-K334A (*rad51*-3A) mutant, which encodes protein that can bind

DNA and form a stable nucleoprotein filament but cannot catalyse DNA strand exchange[53,54], exhibits a twofold reduction in *RTS1*-AO-induced deletions compared to a *rad51*⁺ strain (*p* value = 0.0004) (Supplementary Fig. 2).

Altogether our data show that Rad51-mediated strand invasion is the main driver of gene conversions associated with replication fork blockage and RDR initiation. If we assume that fully active Rad51, in the presence of its auxiliary proteins, is just as strong a barrier to Rad51-independent ectopic recombination as Rad51 that is rendered non-functional by mutation or absence of an auxiliary protein, then we can conclude that Rad51-mediated strand invasion is also the main driver of *RTS1*-AO-induced deletions. However, even in the presence of non-functional Rad51, some *RTS1*-AO-induced deletions are formed by a Rad51-independent pathway. This Rad51-independent ectopic recombination, which increases when Rad51 is absent, may be linked to RDR initiation and/or inter-fork strand annealing during fork convergence[55,56].

## RDR and its associated template switching is only partially dependent on Rad51

Having established that ectopic recombination associated with RDR initiation is mainly driven by Rad51-mediated strand invasion, we next assessed its importance for RDR-associated template switching using strains containing the 12.4 kb reporter (Figs. 1c, d and 3a). As shown previously, *RTS1*-AO strongly induces template switching at this reporter resulting in a ~32-fold increase in deletions (*p* value ≤ 0.0001) and a ~6-fold increase in gene conversions (*p* value ≤ 0.0001)[30,31]. In a *rad51Δ* mutant, the frequency of *RTS1*-AO-

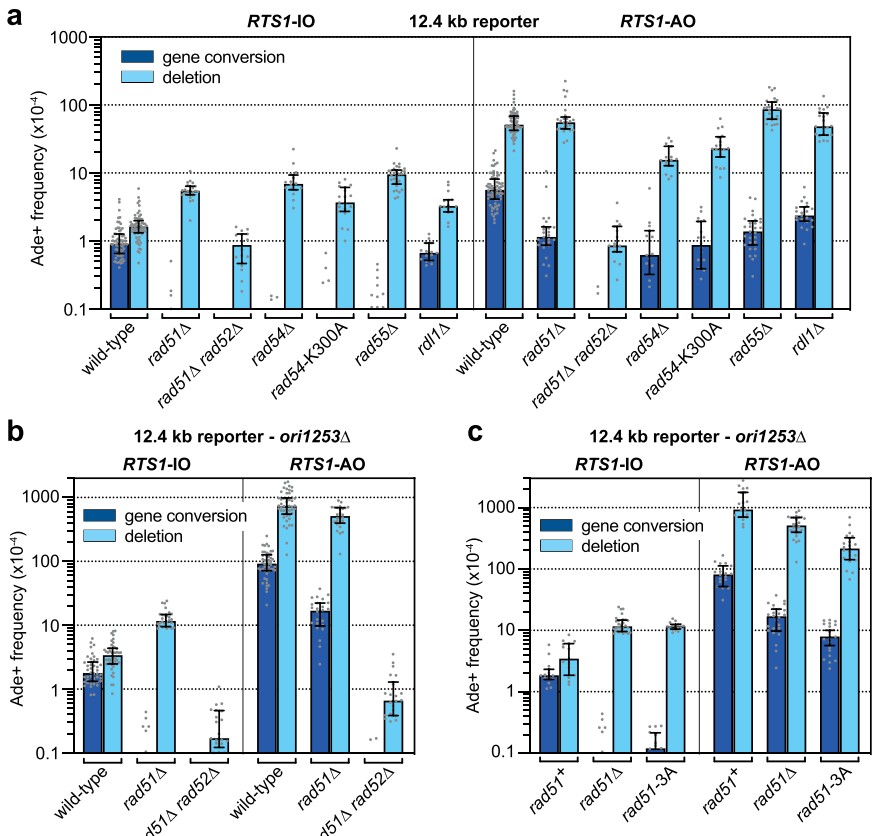

**Fig. 3 | RDR-associated template switching is only partially dependent on Rad51 and Rad51 is only a partial barrier to Rad51-independent template switching. a** Frequency of spontaneous (*RTS1*-IO) and *RTS1*-AO-induced Ade⁺ recombinants in wild-type and mutant strains carrying the 12.4 kb reporter. **b** Frequency of spontaneous (*RTS1*-IO) and *RTS1*-AO-induced Ade⁺ recombinants in wild-type and mutant strains with *ori-1253Δ* and the 12.4 kb reporter. **c** Frequency of spontaneous (*RTS1*-IO) and *RTS1*-AO-induced Ade⁺ recombinants in *rad51⁺*, *rad51Δ*

and *rad51*-3A strains with *ori-1253Δ* and the 12.4 kb reporter. **a–c** Data are presented as median values ± interquartile range with individual data points shown as grey dots. The data are also reported in Supplementary Table 1, which includes the strain numbers, the number of colonies tested for each strain (*n*) and *p* values. Further details of the statistical analysis are reported in Supplementary Data 1. Source data are provided as a Source Data file.

induced gene conversions is reduced by ~5-fold compared to wild-type (*p* value ≤ 0.0001), whereas the frequency of deletions remains at wild-type levels (*p* value ≥ 0.9999) (Fig. 3a). As the amount of template switching at the 12.4 kb reporter is limited by oncoming canonical DNA replication emanating from replication origin *ori-1253* (Fig. 1c)[31], we also assessed template switching in strains where *ori-1253* is deleted (Fig. 3b). In this strain background, *RTS1*-AO induces a ~216-fold increase in deletions and a ~52-fold increase in gene conversions relative to spontaneous levels (*p* values ≤ 0.0001). Deletion of *rad51* in an *ori-1253Δ* background causes a ~5.5-fold reduction in the frequency of *RTS1*-AO-induced gene conversions (*p* value ≤ 0.0001) (Fig. 3b), which is similar to the reduction in the *ori-1253⁺* background. A modest ~1.4-fold reduction in deletions is also detected (*p* value = 0.0175) (Fig. 3b). These data show that RDR-associated template switching that gives rise to deletions is largely independent of Rad51. Even gene conversions, which are strongly dependent on Rad51 during the initiation of RDR, appear to be less dependent on Rad51 when formed by template switching during RDR's elongation phase as indicated by their ~9.5-fold higher level in a *rad51Δ ori-1253Δ* mutant compared to the *ori-1253Δ* spontaneous level (*p* value = 0.0004) (Fig. 3b). The fact that template switching still occurs at relatively high levels in a *rad51Δ* mutant implies that RDR remains active. This Rad51-independent RDR, and associated template switching, is dependent on Rad52 as evidenced by the loss of *RTS1*-AO-induced recombination in a *rad51Δ rad52Δ* double mutant (Fig. 3a, b).

## Rad51 and its binding to DNA is only a partial barrier to Rad51-independent RDR and associated template switching

Our data indicate that Rad52 is capable of driving RDR without Rad51. However, it was unclear whether Rad51-independent RDR only occurs under pathological conditions, when Rad51 is absent, or can function in wild-type cells as an alternative to the Rad51-dependent pathway. To address this question, we first looked at the amount of *RTS1*-AO-induced template switching in Rad51 auxiliary factor mutants, which effectively disable Rad51's strand invasion activity but not its ability to inhibit Rad51-independent ectopic recombination induced by *RTS1*-AO at the 0 kb reporter. Four mutants, *rad54Δ*, *rad54*-K300A, *rad55Δ* and *rdl1Δ*, which had exhibited reduced levels of *RTS1*-AO-induced deletions at the 0 kb reporter, were tested (Fig. 3a). Each one of these mutants exhibits a reduction in *RTS1*-AO-induced gene conversions, ranging from ~2-fold in a *rdl1Δ* mutant (*p* value = 0.0058) to ~4-fold in a *rad55Δ* mutant (*p* value ≤ 0.0001) and ~6–9-fold in *rad54Δ* and *rad51*-K300A mutants (*p* values ≤ 0.0001). *rad54Δ* and *rad51*-K300A mutants also exhibit a ~2–3-fold reduction in *RTS1*-AO-induced deletions (*p* values ≤ 0.0001), which is similar to their impact on ectopic recombination at the 0 kb reporter. In contrast, neither *rad55Δ* nor *rdl1Δ* mutant exhibit a reduction in *RTS1*-AO-induced deletions. If anything, a *rad55Δ* mutant exhibits a ~1.7-fold increase in deletions (*p* value = 0.0039). These data suggest that the Rad51-independent pathway of template switching is not inhibited by Rad51 when Rad55 or Rdl1 are absent. The same may not be true when Rad54 is absent, or rendered defective for its ATPase activity, as *rad54Δ* and *rad54*-K300A mutants

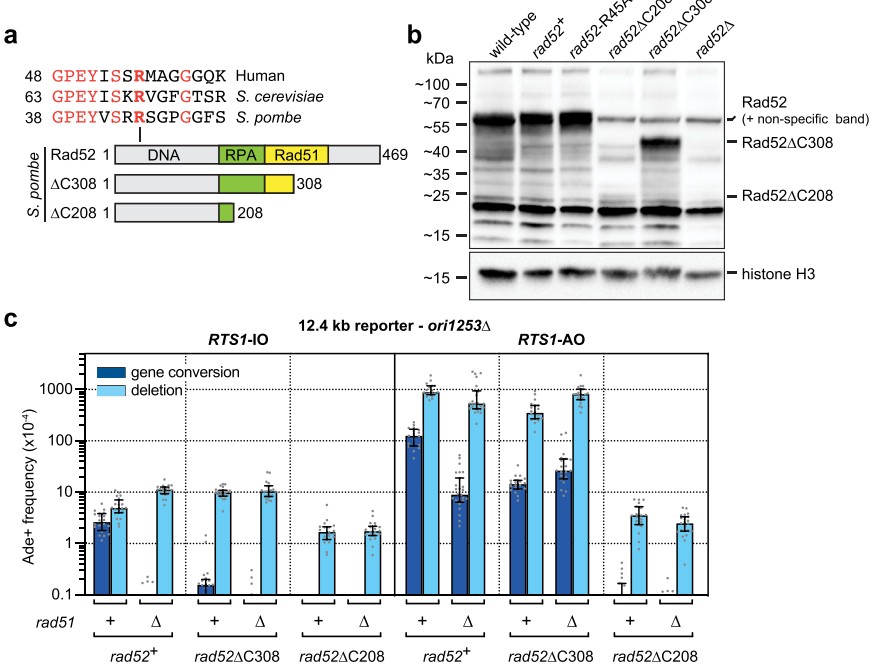

**Fig. 4 | Rad52's N-terminal DNA binding domain is required for Rad51-independent template switching. a** Schematic showing the main interaction domains and conserved arginine-45 in fission yeast Rad52. The two C-terminal truncations, ΔC308 and ΔC208, are also shown. **b** Western blot showing the relative amounts of full length and truncated Rad52 in whole cell extracts from wild-type (MCW8888), *rad52⁺-kanMX6* (MCW9694), *rad52*-R45A-*kanMX6* (MCW9840), *rad52*ΔC208-*kanMX6* (MCW9692), *rad52*ΔC308-*kanMX6* (MCW9693) and *rad52*Δ (MCW1285) strains. Histone H3 serves as a loading control. An independent repeat of this experiment gave similar results. **c** Frequency of spontaneous (*RTS1*-IO) and *RTS1*-AO-induced Ade⁺ recombinants in *rad52⁺* and *rad52* C-terminal truncation mutant strains with *ori1253*Δ and the 12.4 kb reporter. The data are also reported in Supplementary Table 1, which includes the strain numbers, the number of colonies tested for each strain (*n*) and *p* values. Further details of the statistical analysis are reported in Supplementary Data 1. Source data are provided as a Source Data file.

exhibit ~2–3-fold lower frequency of *RTS1*-AO-induced deletions compared to a *rad51*Δ mutant (*p* values ≤ 0.0004) (Fig. 3a). However, this lower level of deletions may not reflect inhibition of the Rad51-independent pathway, it could instead indicate that Rad54 is required to promote it.

To determine more directly whether Rad51 inhibits Rad51-independent RDR, we measured template switching at the 12.4 kb reporter in a *rad51*-3A mutant. As mentioned above, Rad51-3A can bind DNA and form a stable nucleoprotein filament but cannot catalyse DNA strand exchange[53,54]. Consistent with Rad51-3A being defective for strand exchange, the *rad51*-3A mutant exhibits a > 15-fold reduction in spontaneous gene conversions, and a ~3-fold increase in spontaneous deletions, similar to a *rad51*Δ mutant (Fig. 3c). With *RTS1*-AO, a *rad51*-3A mutant exhibits ~2-fold lower levels of deletions compared to a *rad51*Δ mutant (*p* value = 0.0016) (Fig. 3c). The value for gene conversions is also ~2-fold lower but this difference is not statistically significant (*p* value = 0.0610). However, even though *RTS1*-AO-induced template switching is reduced in a *rad51*-3A mutant, gene conversions are still ~4-fold higher and deletions ~62-fold higher than wild-type spontaneous levels (*p* values ≤ 0.0001). Altogether, these data suggest that the presence of Rad51, and its binding to DNA, only partially inhibits Rad51-independent RDR and associated template switching.

## Rad52's N-terminal DNA binding domain is required for Rad51-independent template switching

Having established that Rad51-independent RDR and associated template switching is active in fission yeast and can function in the presence of Rad51, we next sought to determine which properties of Rad52 are required for this pathway. Specifically, we investigated whether Rad52's C-terminal disordered domain, and RPA and Rad51 binding domains are required by testing C-terminal truncations of Rad52, which lack these domains (Fig. 4a). A *rad52*ΔC308 mutant,

which lacks the C-terminal disordered domain and most of the Rad51 binding domain, exhibits a similar phenotype as a *rad51*Δ single mutant for both spontaneous recombination and *RTS1*-AO-induced template switching and the interaction between the two mutants is epistatic (Fig. 4c and Supplementary Table 1). In contrast, a *rad52*ΔC208 mutant, which additionally lacks Rad52's RPA binding domain, exhibits relatively little or no *RTS1*-AO-induced Ade⁺ recombinants implying an almost complete deficit in RDR and/or template switching (Fig. 4c). However, unlike Rad52ΔC308, which accumulates to similar levels as full-length protein, very little Rad52ΔC208 is detected by western blotting suggesting that this truncated form of Rad52 is either poorly expressed or unstable (Fig. 4b). To overcome the problem of poor expression or protein instability, we tried over-expressing a similar truncated form of Rad52 (Rad52ΔC210) from the *nmt41* promoter in plasmid pREP41 (Supplementary Fig. 3a). Over-expressed Rad52ΔC210 accumulated to similar levels as endogenously expressed full-length Rad52 but much lower levels than full-length Rad52 and Rad52ΔC316 expressed from the same plasmid-borne promoter (Supplementary Fig. 3b). Overexpression of Rad52 or Rad52ΔC316 had little or no effect on the frequency of spontaneous and *RTS1*-AO-induced Ade⁺ recombinants in wild-type and *rad51*Δ strains, but rescued recombination in a *rad51*Δ *rad52*Δ double mutant to near *rad51*Δ mutant levels (Supplementary Fig. 3c, d). Similar results were observed with Rad52ΔC210, although the rescue of *RTS1*-AO-induced Ade⁺ recombinants in *rad51*Δ *rad52*Δ mutant cells was less than with full-length Rad52 or Rad52ΔC316 (*p* values = 0.0116–0.0001) (Supplementary Fig. 3c, d). Altogether these data suggest that Rad52's N-terminal DNA binding domain is sufficient to drive Rad51-independent RDR and associated template switching, albeit inclusion of its RPA binding domain enhances this ability, at least in part, by promoting its expression and/or stability.

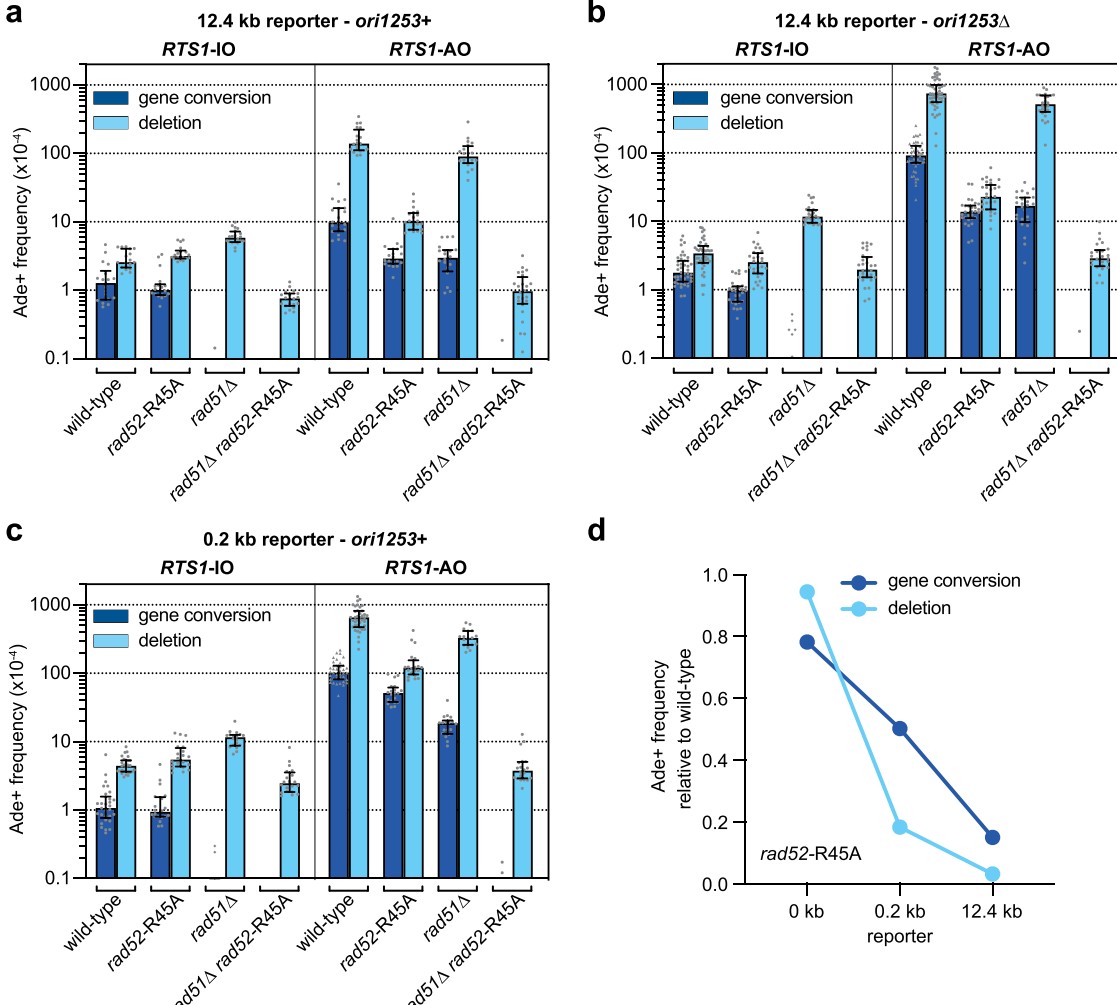

**Fig. 5 | Rad52's DNA annealing activity is required for RDR-associated template switching. a** Frequency of spontaneous (*RTS1*-IO) and *RTS1*-AO-induced Ade⁺ recombinants in *ori-1253*⁺ strains carrying the 12.4 kb reporter. **b** Frequency of spontaneous (*RTS1*-IO) and *RTS1*-AO-induced Ade⁺ recombinants in *ori-1253*Δ strains carrying the 12.4 kb reporter. **c** Frequency of spontaneous (*RTS1*-IO) and *RTS1*-AO-induced Ade⁺ recombinants in *ori-1253*⁺ strains carrying the 0.2 kb reporter. **a–c** Data are presented as median values ± interquartile range with individual data points shown as grey dots. The data are also reported in Supplementary Table 1, which includes the strain numbers, the number of colonies tested for each strain (*n*) and *p* values. Further details of the statistical analysis are reported in Supplementary Data 1. Source data are provided as a Source Data file. **d** Frequency of *RTS1*-AO-induced Ade⁺ recombinants in a *rad52*-R45A mutant relative to wild-type. Values are derived from the data in (**a–c**).

## Rad52's DNA annealing activity is required for RDR-associated template switching

We next sought to confirm that Rad52's DNA annealing activity is required for Rad51-independent RDR and associated template switching by testing the *rad52*-R45A mutant described earlier (Fig. 4a, b). Rad51-independent *RTS1*-AO-induced template switching at the 12.4 kb reporter was abolished in a *rad51*Δ *rad52*-R45A double mutant indicating its reliance on Rad52's DNA annealing activity (Fig. 5a, b). Surprisingly, the frequency of *RTS1*-AO-induced Ade⁺ recombinants was also markedly reduced in a *rad52*-R45A single mutant (by ~11-fold in an *ori-1253*+ strain [*p* value ≤ 0.0001] and ~21-fold in an *ori-1253*Δ strain [*p* value ≤ 0.0001]) (Fig. 5a, b). This result is in stark contrast with the little or no effect that it has on spontaneous recombination and *RTS1*-AO-induced recombination at the 0 kb reporter (Fig. 2a, b), and indicates that Rad52's DNA annealing activity is required for RDR-associated template switching even when Rad51 is present. To further evaluate this, we assessed template switching 0.2 kb downstream of *RTS1* (0.2 kb reporter) (Fig. 5c). As with the 12.4 kb reporter, the frequency of Ade⁺ recombinants was reduced in the *rad52*-R45A mutant, however, the fold reduction (~4-fold) was markedly less. A comparison of the effect of the *rad52*-R45A mutation on *RTS1*-AO-induced recombination at the 0, 0.2 and 12.4 kb reporters reveals an increasing dependence on Rad52's DNA annealing activity as RDR progresses from the site of replication fork collapse (Fig. 5d).

## Rad52's DNA annealing activity drives template switching between diverged *Alu* elements

So far, we have focused on measuring template switching between almost perfect copies of the ~1.7 kb *ade6* gene. However, in humans, many disease-associated genomic rearrangements, thought to be caused by template switching, occur between relatively short stretches of homeologous or microhomologous DNA[24]. For example, template switching between divergent *Alu* elements, which constitute ~11% of the human genome[57], is thought to account for genomic deletions associated with hereditary spastic paraplegia[58–60]. To see if RDR-associated template switching can occur between divergent *Alu* elements, and assess whether Rad52's DNA annealing activity is required for this, we constructed a new genetic reporter consisting of a direct repeat of *Alu*Sx1 and *Alu*Sp, separated by ~3.8 kb of DNA containing *ura4*⁺ and *his3*⁺ genes, and positioned 12.4 kb

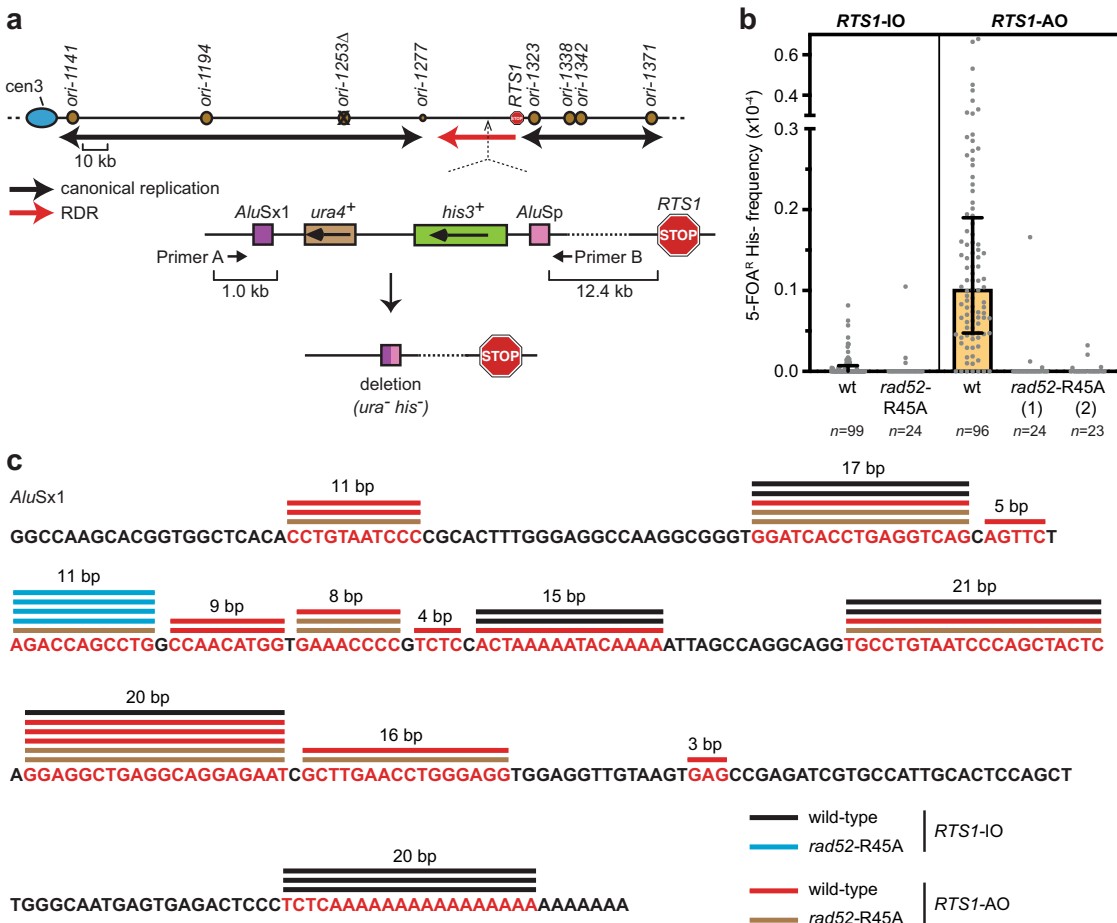

**Fig. 6 | Rad52's DNA annealing activity drives template switching between diverged *Alu* elements. a** Schematic showing the *Alu*Sx1-*Alu*Sp recombination reporter inserted 12.4 kb downstream of *RTS1* on chromosome 3. The upper part of the panel shows a map of the chromosome 3 region where *RTS1* and the *Alu*Sx1-*Alu*Sp recombination reporter are inserted. Replication origins are indicated by brown circles. Black and red arrows indicate the direction of canonical replication and RDR, respectively. The lower part of the panel shows the *Alu*Sx1-*Alu*Sp recombination reporter. Primers A and B are used to amplify the *Alu*Sx1-*Alu*Sp region for analysis of recombinant products. **b** Frequency of spontaneous (*RTS1*-IO) and *RTS1*-AO-induced Ura⁻ His⁻ recombinants in wild-type and *rad52*-R45A mutant strains with *ori-1253*Δ and the *Alu*Sx1-*Alu*Sp recombination reporter shown

in (**a**). Data are presented as median values ± interquartile range with individual data points shown as grey dots. *n* indicates the number of colonies tested for each strain. Each data point represents the frequency of Ura⁻ His⁻ recombinants in one of the tested colonies. The data are also reported in Supplementary Table 2, which includes the strain numbers and *p* values. Further details of the statistical analysis are reported in Supplementary Data 1. Source data are provided as a Source Data file. **c** Sequence analysis of *Alu*Sx1-*Alu*Sp template switches. The sequence of *Alu*Sx1 is shown with homologous sequences used for template switching in red. The coloured lines above indicate the number of events observed among the subset of colonies tested for each strain.

downstream of *RTS1*-IO/AO on chromosome 3 (Fig. 6a). *Alu*Sx1 and *Alu*Sp share 88.2% identity and are capable of mediating template switching associated with BIR in budding yeast and disease-associated deletions in humans[23,60]. In our assay, template switching between the ~300 bp *Alu* elements results in deletion of the *ura4⁺* and *his3⁺* genes (Fig. 6a). As loss of *ura4* confers resistance to 5-fluoroorotic acid (5-FOA), the frequency of deletion events can be measured by plating cells on 5-FOA followed by replica plating onto minimal media lacking histidine. With *RTS1*-IO, relatively few colonies (33 out of 99) contained 5-FOA resistant His⁻ cells resulting in a median frequency of zero (Fig. 6b). In contrast, almost all (89 out of 96) colonies with *RTS1*-AO contained 5-FOA resistant His⁻ cells with an overall median frequency of ~0.1 × 10⁻⁴ (*p* value ≤ 0.0001) (Fig. 6b). PCR and sequence analysis of a random selection of 5-FOA resistant His⁻ colonies confirmed that the deletions were formed by *Alu*Sx1-*Alu*Sp template switches at sites of microhomology ranging from 3 to 21 base pairs (Fig. 6c). These data show that RDR-associated template switching between divergent *Alu* elements can generate genomic deletions. Similar to template switching between *ade6* alleles, *RTS1*-AO-induced *Alu*Sx1-*Alu*Sp template switches are abolished in a

*rad52*-R45A mutant indicating that Rad52's DNA annealing activity is required (Fig. 6b).

## Discussion

### Both Rad51-dependent and Rad51-independent pathways drive RDR in fission yeast

Pioneering studies in budding yeast have established that there are two pathways of BIR: a major pathway that depends on Rad51 and Rad52; and a minor pathway that depends on Rad52 but is independent of Rad51[4,13]. We have shown here that RDR triggered by fork collapse in fission yeast also occurs via Rad51-dependent and Rad51-independent pathways. If we assume that the frequency of template switching is an accurate indicator of the amount of RDR, then both pathways appear to hold more equal sway in driving RDR in fission yeast than they do for BIR in budding yeast. Indeed, by comparing the frequency of template switching at the 12.4 kb reporter in wild-type and *rad51*-3A cells, we can estimate that ~22% of total RDR can be initiated and driven by the Rad51-independent pathway when Rad51 is present and capable of binding DNA. However, we cannot rule out the possibility that template switching is a poor indicator of overall RDR levels, as it might

have its own specific genetic requirements and occur in only a minority of RDR events. Therefore, the contribution that the Rad51-independent pathway makes to total RDR remains uncertain. Nevertheless, at least for RDR that is associated with template switching, the Rad51-independent pathway is active and able to make a significant contribution to its initiation and elongation even when Rad51 is present. When Rad51 is absent, the frequency of template switch recombination at the 12.4 kb reporter is ~2-fold higher than in a *rad51*-3A mutant (*p* value = 0.0016), which suggests that the Rad51 nucleofilament is at least a partial barrier to the Rad51-independent pathway. Whether this is sufficient to restrict the Rad51-independent pathway to the period prior to Rad51 nucleofilament formation remains an open question.

### Transitioning from Rad51-mediated strand invasion to Rad52 DNA annealing during RDR

Once initiated, RDR must suffer from subsequent collapse events where the newly synthesised DNA strand disengages from its template. Restarting RDR, following such disengagement, presumably requires the nascent strand to re-invade the template with misguided re-invasions resulting in template switches[30,31]. At the site of fork collapse most ectopic recombination is driven by Rad51 and there is no reliance on Rad52's DNA annealing activity. In contrast, ~77% of template switch recombination depends on Rad52's DNA annealing activity 0.2 kb downstream of the site of fork collapse and this increases to ~96% at 12.4 kb downstream. This discovery, that template switching becomes increasingly reliant on Rad52's DNA annealing activity the further RDR progresses from its initiation site, could be explained by a model in which the Rad51-independent pathway drives RDR elongation at a faster pace than the Rad51-dependent pathway, with both pathways remaining completely separate. However, we favour an alternative model in which the catalysis of strand invasion is driven mainly by Rad51 during RDR initiation and then transitions to Rad52 during RDR elongation. A similar transition was recently reported for alternative lengthening of telomeres (ALT) in telomerase defective budding yeast[61]. Like RDR, ALT is driven by Rad51-dependent and Rad51-independent pathways that generate so-called type I and type II survivors respectively. It was thought that these pathways functioned independently of each other, however Malkova and colleagues showed that they instead work together to forge hybrid type I/type II chromosome ends through a unified pathway initiated by Rad51-mediated strand invasion, and completed by a Rad51-independent mechanism that relies on Rad52 and Rad59[61]. The proposed transition from Rad51-dependent to Rad51-independent RDR is also reminiscent of Haber and colleagues' finding that template switching, associated with Rad51-dependent BIR in budding yeast, exhibits a reduced requirement for sequence homology and reliance on Rad51 compared to the strand invasion that initiates the BIR process[22]. Whilst they did not investigate whether Rad52's DNA annealing activity is required they did show that template switching depends on Rdh54 even though it is unnecessary for the initial strand invasion[22]. As Rdh54 is required for Rad51-independent BIR, this suggests that BIR, like *RTS1*-AO-induced RDR, may transition from an initiation event driven by Rad51-mediated DNA strand invasion to an elongation phase where Rad52-mediated DNA annealing promotes template switching. However, it is important to note that BIR can progress more than 100 kb in budding yeast with seemingly only a small reduction in efficiency when Rad52's strand annealing activity is absent[62]. Therefore, any transition from Rad51-dependent to Rad51-independent pathway is not essential for the elongation phase of BIR. How important it is for RDR in fission yeast remains to be determined.

What causes the proposed transition from Rad51-dependent to Rad51-independent template switching remains unclear. One possibility is slow re-cycling of Rad51 and/or its auxiliary proteins from the site of RDR initiation, which could lead to their localised exhaustion.

Alternatively, the activity of Rad51 disruptases, such as Fbh1 and Srs2 that are known to suppress template switching[30], might be upregulated. Or, Rad52's DNA annealing activity may be activated in preference to its Rad51 mediator function. Whatever the cause, determining how the transition occurs is an important goal for future research. It will also be important to determine whether there is any advantage to the cell in switching from using Rad51 to Rad52 during RDR's elongation phase. Rad52 DNA annealing may be a faster and more efficient way of re-engaging the nascent strand following dissociation from its template enabling RDR to progress more rapidly. Proper convergence of the canonical replication fork with RDR presumably requires the RDR nascent strand to be annealed to its template. Therefore, limiting the amount of time that the nascent strand remains in a dissociated state would also reduce the chance of the oncoming replication fork migrating past it and creating an over-replication problem. Using Rad52 to keep the nascent strand engaged with its template may be an efficient way of perpetuating RDR and preventing over-replication but, as its requirement for DNA homology is less stringent than Rad51's, it presumably comes with a greater risk of template switching[14,63,64].

### How does Rad52 initiate RDR and catalyse template switching?

We have shown that Rad52's N-terminal DNA binding domain is sufficient to drive RDR and its associated template switching in the absence of Rad51. This domain can promote annealing of complementary ssDNAs and generates D-loops between ssDNA and supercoiled plasmid DNA[65–68]. However, it remains unclear whether its D-loop forming activity works by a Rad51-type strand invasion mechanism, or simply by annealing the ssDNA to patches of ssDNA within the plasmid[68]. In vivo, Rad51-independent BIR is thought to rely on Rad52 annealing the resected end of a DSB to homologous or homeologous ssDNA exposed at a replication fork, transcription complex or secondary DNA structure[15]. Rad51-independent RDR in fission yeast may similarly depend on such chance exposed stretches of ssDNA. Alternatively, there may be proteins that work with Rad52 to promote its D-loop forming activity by transiently unwinding DNA. Determining the exact mechanism by which Rad52 initiates RDR and promotes template switching remains an important goal for future research.

### Rad51-independent RDR and Rad51 paralogues

We have shown that Rad51 can partially inhibit Rad52-dependent *RTS1*-AO-induced ectopic recombination at the 0 kb reporter when its proper nucleofilament formation and activity are impaired by deletion of key auxiliary proteins such as Rad55, Rad57, Rdl1 and Rlp1. However, unlike in a *rad51*-3A mutant, this inhibition does not correlate with a similar reduction in *RTS1*-AO-induced template switching at the 12.4 kb reporter. Indeed, there is little or no overall reduction in template switch recombination at the 12.4 kb reporter in either a *rad55Δ* or *rdl1Δ* mutant. How can this be if both RDR pathways are impaired? Possibly, the frequency of template switching increases in these mutants, which compensates for the reduction in RDR. As Rad55 interacts with Rad52 in both budding and fission yeast, it could directly inhibit Rad52 mediated template switching, together with its partner Rad57 and the Rdl1-Rlp1-Sws1 complex[47,69]. Alternatively, RDR initiation may remain at wild-type levels whilst ectopic recombination is impaired. This could happen if the DNA used for the homology search is confined to the immediate vicinity of the collapsed fork where the absence of *ade6* sequence would preclude ectopic recombination in our assay. Without Rad55-Rad57 or Rdl1-Rlp1-Sws1, the assembly of the Rad51 nucleofilament is likely to be suboptimal but not completely abolished[47,70,71]. This suboptimal nucleofilament may either be sufficient to initiate RDR (but not ectopic recombination) itself or able to restrict Rad52's DNA annealing activity to the 3′ end of the reversed fork where it could initiate RDR without generating Ade+ recombinants.

### Relevance to human disease

It has long been suspected that BIR can transition from a relatively accurate initiation phase driven by Rad51-mediated strand invasion to a much less accurate elongation phase where template switching between homeologous and microhomologous DNA is rife[22,28,29]. Indeed, it is thought that this transition is instrumental in the genesis of many complex genome rearrangements, including chromothripsis, that are found in congenital diseases and cancers[4]. Our discovery that the repair of a collapsed replication fork in fission yeast appears to involve a switch from Rad51-dependent RDR to Rad51-independent RDR, provides support for the transition hypothesis. Given that both RDR pathways seem to be conserved in humans[17,18,72], and human and fission yeast Rad52 can promote DNA annealing between micro-homologous sequences[63,64], we suspect that the transition we observe in fission yeast also occurs in humans and likely accounts for many disease-associated genomic rearrangements.

## Methods

### Yeast strains and plasmid construction

*S. pombe* strains are listed in Supplementary Table 3. Derivatives of recombination reporter strains carrying the indicated gene/replication origin deletion(s) were obtained from genetic crosses. The 12.4 kb reporter strains marked with *ura4*[+] were obtained by a one-step marker swap protocol in which *kanMX6* in MCW7257/MCW7259 was replaced by *ura4MX4*[73]. To replace *rad52* with *rad52*[+]-*kanMX6* or *rad52*-R45A-*kanMX6*, a derivative of pFA6a-*kanMX6*[74] (pEB8) containing the *rad52* 5′ untranslated region (UTR), open reading frame (ORF) and 3′ UTR was first constructed. pEB6 was then derived from pEB8 by site-directed mutagenesis, using primers oMW2023 (CGTTTCAAG AGCGTCAGGTCCTGG) and oMW2024 (TACTCAGGTCCAAGCTTTC), to create an AG to GC substitution at nucleotides 230–231 in the *rad52* ORF. The *rad52*[+]-*kanMX6* and *rad52*-R45A-*kanMX6* gene targeting cassettes were then excised from pEB8 and pEB6, respectively, by digesting with SalI and EcoRI, and transformed into FO652 by a lithium acetate protocol[75]. Strains with *rad52ΔC208*-*kanMX6* and *rad52ΔC308*-*kanMX6* were derived from YSK194 and YSK212, respectively[40]. First, gene targeting cassettes encompassing *rad52ΔC208*-*kanMX6* and *rad52ΔC308*-*kanMX6* were amplified from YSK194 and YSK212 genomic DNA, using primers oMW2020 (CTTTCATAAAAAAACAGAAAAC AT) and oMW2046 (GGATGAATGTCTGCGACT). These cassettes were then transformed into FO652. To replace *rad54* with *rad54*-K300A-*natMX4*, a derivative of pAG25[76] (pCB22), containing the *rad54* 5′ UTR, ORF and 3′ UTR plus *ADH1* terminator, was constructed. pCB22 was then subjected to site-directed mutagenesis, using primers oMW1571 (CAGATGAGATGGGACTTGGTGCGACACTTCAATGTATTGCTT) and oMW1572 (AAGCAATACATTGAAGTGTCGCACCAAGTCCCATCTCATC TG), to create an AA to GC substitution at nucleotides 898–899 in the *rad54* ORF. The resulting plasmid (pCB28) was digested with XbaI and SpeI and the liberated gene targeting cassette transformed into FO652 by a lithium acetate protocol[75]. To replace *rad51* with *rad51*[+]-*kanMX6* or *rad51*-3A-*kanMX6*, gene targeting constructs (pEB2 and pEB1) were made by inserting *rad51* 5′ UTR, ORF and 3′ UTR sequences into pFA6a-*kanMX6*[74]. The *rad51*-3A DNA used to make pEB1 was derived from the yeast strain AA133[53]. The *rad51*[+]-*kanMX6* and *rad51*-3A-*kanMX6* gene targeting cassettes were excised from pEB2 and pEB1, respectively, by digestion with SalI and SapI, and then transformed into FO652 by a lithium acetate protocol[75]. In each of the above cases, diagnostic PCRs and DNA sequencing were used to confirm the successful replacement of the wild-type gene. This analysis also revealed that the *rad52* truncations, *rad52ΔC208*-*kanMX6* and *rad52ΔC308*-*kanMX6*, were slightly different than reported for YSK194 and YSK212[40].

Plasmids for the overexpression of Rad52 (pMW613), Rad52ΔC316 (pAK5) and Rad52ΔC210 (pAK13) were made by first amplifying the appropriate DNA fragments from FO652 genomic DNA using primers oMW551 (TTATACATATGTCTTTTGAGCAAAAACAGCATG) plus oMW

552 (TAGGATCCTTATCCTTTTTTGGCTTTCTTATCCAC) (Rad52), oMW 551 plus oMW1950 (ATGGATCCTTATTAAGTCACAGCATTGCCAACG) (Rad52ΔC316) and oMW551 plus oMW2022 (ATGGATCCTTATTATTG ATTGTTAACAGTCCTCG) (Rad52ΔC210), and then cloning these as NdeI-BamHI fragments into pREP41[77]. Plasmids were verified by DNA sequencing.

To make the *Alu*Sx1 – *Alu*Sp genetic reporter, a DNA construct containing the *Alu*Sx1 and *Alu*Sp DNA sequences[23], with intervening restriction sites and flanking sequences from the 5′ to 3′ ends of *ade6*, was synthesised. DNA fragments containing the *ura4* and *his3* genes were then subcloned into the interval between *Alu*Sx1 and *Alu*Sp. The resulting plasmid (pMW935) was digested with NotI to liberate a gene targeting cassette containing the *Alu*Sx1 – *Alu*Sp genetic reporter, which was transformed into MCW9093 and MCW9094 by a lithium acetate protocol to derive strains MCW9105 and MCW9109, respectively. The *RTS1*-IO/AO-*hphMX4* sequences in MCW9105 and MCW9109 were then exchanged to *RTS1*-IO/AO-*LEU2* by a marker swap protocol using NotI digested pMW939 and pMW938, which are derivatives of pMW922 and pMW921, respectively[31].

### Media and growth conditions

Standard protocols were used for the growth and genetic manipulation of *S. pombe*[78]. The complete and minimal media were yeast extract with supplements (YES) and Edinburgh minimal medium plus 3.7 g/l sodium glutamate (EMMG) and appropriate amino acids (225 mg/l), respectively. Strains carrying pREP41 or its derivatives were grown on EMMG lacking leucine and supplemented with thiamine (4 μM final concentration) where appropriate. To maintain the integrity of the various genetic reporters, strains carrying them were grown on EMMG lacking histidine and/or uracil as appropriate. In addition, strains carrying *ade6-L469* – *ade6-M375* genetic reporters were grown on media supplemented with low levels of adenine (10 mg/l) to distinguish non-recombinant colonies (red) from Ade[+] recombinants (white). Ade[+] recombinants were selected on YES lacking adenine and supplemented with 200 mg/l of guanine to prevent uptake of residual adenine (YE-ade+gua). Ura[−] recombinants were selected on YES containing 1.5 g/l of 5-FOA. All strains were grown at 30 °C.

### Recombination assays

The protocol for determining the frequency of Ade[+] recombinants has been described[79]. In brief, strains were grown for 4–5 days on YES plates at 30 °C, except for the data in Supplementary Fig. 3c and d where strains were grown for 6–7 days on EMMG plates lacking leucine and thiamine to select for plasmids and allow for expression of full-length and truncated Rad52. Similar sized initial colonies were then suspended in 1 ml of water and serially diluted. Appropriate dilutions were plated on YES with low levels of adenine (10 mg/l) (YES/LA) and YE-ade+gua plates. Colonies were counted after 4–6 days incubation at 30 °C using a Protos 3 automated colony counter (Synbiosis), and the percentage of Ade[+] colonies amongst total viable (colony forming) cells was determined by comparing the number of colonies on the YES/LA and YE-ade+gua plates. The proportion of deletions (Ade[+] His[−]) and gene conversions (Ade[+] His[+]) was determined by replica plating the YE-ade+gua plates onto EMMG plates lacking histidine. Each strain was assayed at least twice with between 5 and 20 colonies analysed in each experiment. Recombination frequencies were determined using the method of the median to prevent skewing of the data by jackpot events, where a single recombination event at an early stage in the growth of the colony can give rise to many recombinant cells.

The protocol for determining the frequency of 5-FOA resistant colonies in strains containing the *Alu*Sx1 – *Alu*Sp genetic reporter is modified from Osman and Whitby[79]. Strains were grown for 4–4.5 days on YES plates. Similar sized initial colonies were then suspended in 0.2 ml of water and serially diluted. Appropriate dilutions were plated on YES and YES plus 5-FOA plates. Colonies were counted after 4 days

incubation at 30 °C using a Protos 3 automated colony counter (Synbiosis), and the percentage of 5-FOA resistant colonies amongst total viable (colony forming) cells was determined by comparing the number of colonies on the YES and YES plus 5-FOA plates. The percentage of His⁻ 5-FOA resistant colonies was determined by replica plating the YES plus 5-FOA plates onto EMMG plates lacking histidine. Each strain was assayed at least three times with 8 or more initial colonies analysed in each experiment. Recombination frequencies were determined using the method of the median. Breakpoint analysis was conducted on a random selection of His⁻ 5-FOA resistant colonies, with no more than one colony from each initial colony being analysed. This analysis involved PCR amplification from the His⁻ 5-FOA resistant colony using Primers A (oMW1990: TGCATCATTTACTGACCCCG) and B (oMW1991: TGGCTATTATTGATAGCAACAG), which flank the *Alu*Sx1 – *Alu*Sp genetic reporter (Fig. 6a), followed by DNA sequencing of the PCR product.

### Statistical analysis

Analysis of recombination data was performed using Excel for Mac Version 16.66.1 (Microsoft®) and GraphPad Prism Version 9.3.1 (GraphPad Software, San Diego, CA). Recombination frequencies were analysed for normal distribution using the Shapiro-Wilk test. Not all data passed this test and, therefore, recombination frequencies were compared using a Kruskal–Wallis test (one-way ANOVA on ranks) with a Dunn's multiple comparisons post-test or a two-tailed Mann–Whitney test. These are non-parametric statistical tests and, therefore, do not require the data to be normally distributed. *P* values are reported in the main text, Supplementary Tables 1 and 2, and in Supplementary Data 1.

### Spot assay

Cells growing exponentially in YES at 30 °C were harvested, washed, counted using a haemocytometer and resuspended in water at a density of $5 \times 10^6$ cells/ml. The suspension was serially diluted in ten-fold steps to $5 \times 10^3$ cells/ml, and 10 µl aliquots of each suspension were spotted onto a series of YES plates with and without genotoxins as indicated. For UV treatment, plates were irradiated using a Stratalinker (Stratagene). The plates were photographed after 4 days at 30 °C.

### Western blotting

Strains were grown in 10 ml cultures to a cell density of $1 \times 10^6$ cells/ml. Cells were then harvested by centrifugation, washed with water, and resuspended in 300 µl of water. An equal volume of 0.6 M NaOH was then added to the cell suspension, which was then incubated at room temperature for 5 min. The cells were then pelleted by centrifugation and lysed by resuspending in 100 µl of SDS loading dye and boiling for 5 min. The cell debris was then removed by centrifugation and the supernatant was run on a SDS-PAGE gel. Proteins were then transferred onto Immun-Blot® PVDF Membrane (Bio-Rad Laboratories, Inc.) and target proteins detected using rabbit Anti-Rhp51/Rad51 (1 in 2000 dilution) (Cosmo Bio Ltd, BAM-63-001-EX), Anti-Rad22/Rad52 (1 in 2000 dilution) (Cosmo Bio Ltd, BAM-63-003-EX) and Anti-Histone H3 (1 in 2000 dilution) (Abcam plc, ab1791) polyclonal antibodies as indicated. The secondary antibody was Anti-Rabbit IgG (whole molecule)-Peroxidase antibody produced in goat (1 in 10000 dilution) (Sigma-Aldrich, A6154), and was detected using Immobilon Crescendo Western HRP substrate (Merck).

### Reporting summary

Further information on research design is available in the Nature Portfolio Reporting Summary linked to this article.

## Data availability

All data generated or analysed during this study are included in this published paper and its Supplementary Information files. The replication origins indicated in Figs. 1c and 6a are listed in OriDB (pombe.oridb.org). Source data are provided with this paper.

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

## Acknowledgements

We thank Sarah Lambert and Grzegorz Ira for the gift of strains, and Claire Bryer and Steven Pilley for constructing some of the plasmids/strains used in this study. This work was supported by grants MR/P028292/1 (awarded to M.C.W.) and MR/V009214/1 (awarded to M.C.W.) from the Medical Research Council, BB/P019706/1 (awarded to M.C.W.) from the Biotechnology and Biological Sciences Research Council, and 090767/Z/09/Z (awarded to M.C.W.) from the Wellcome Trust.

## Author contributions

A.K., S.T., M.O.N., J.O. and M.J.: conceived and performed experiments. E.B. and C.A. performed experiments; C.A.M. and F.O. provided reagents, expertise and feedback; M.C.W. conceived experiments, wrote the paper, and secured funding.

## Competing interests

The authors declare no competing interests.
