## [Peer Review File · Nature Communications]

Rad52's DNA annealing activity drives template switching associated with restarted DNA replicationREVIEWER COMMENTS

Reviewer #1 (Remarks to the Author):

New work by Whitby lab investigates the role of Rad51 recombinase and Rad52 mediated annealing in the recombination dependent replication (RDR) initiated at collapsed replication forks. They used a previously published experimental model in fission yeast to achieve this goal. The readout of the repair is the frequency of template switching leading to gene conversion or deletions. Template switching reporter is inserted either right next to the replication fork barrier where the collapse occurs or 12 kb away from it. This way, they can measure template switches during the initial strand invasion step or RDR elongation step.

The most interesting result of this work is that template switches during the elongation step are independent of Rad51 and dependent on Rad52. The Rad52 domain needed is an annealing domain as demonstrated by using separation of function mutant rad52-R45A. Template switching close to the strand invasion site depends on Rad51 recombinase, and the role of Rad52-annealing is revealed only in the absence of Rad51. The authors propose that RDR starts by Rad51 mediated pathway, but at least some events can be completed by a Rad51-independent pathway. This is an interesting idea as previously recombination events were considered to be either Rad51-dependent or -independent. They also demonstrate that similar template switches can occur between highly diverged human ALU repeats and depend on the annealing activity of Rad52. Finally, they show the minor role of fission yeast SHU complex in RDR and that Rad51 may inhibit the Rad52-dependent mechanism.

Comments

1. The idea that some RDR or BIR events start by the Rad51 pathway but are finished by Rad52-dependent annealing is not novel. Recent work shows that telomere recombination in telomerase deficient cells in budding yeast starts by Rad51 pathway and finishes by mechanism employing Rad52 annealing. This work should be discussed (PMID: 33639094). Despite the similarity of the central conclusion, the work here is important because the recombination type studied is entirely different. It shows that the hybrid mechanism of Rad51-dependent and -independent pathways can likely be applied to different repair contexts.

2. How does annealing mediate template switch during the elongation phase of RDR? Does the distant ade6-L469 repeat have to be transcribed for the annealing? Would a stronger promoter increase annealing? It is not clear if Alu repeats are transcribed. If not transcription, then how does the second repeat of the reporter become single-stranded? Alternatively, if repair going toward and through the reporters is essential for some supercoiling or unwinding that facilitates annealing, would the repeat located on another chromosome be used for template switching in an annealing-independent manner?

3. The authors propose that during RDR, there is a transition from Rad51-dependent to Rad51-independent pathway. However, they mostly measure template switches close or far from strand invasion. Perhaps template switches occur only at a small fraction of events, and RDR is mostly Rad52 annealing independent. Regular BIR in budding yeast where the synthesis goes for over 100 kb is nearly entirely Rad52-annealing independent (PMID: 33844333). However, template switches were not tested. This work should be discussed.

Minor

Line 82. BIR is mutagenic mainly because of the inability to use MMR – conservative inheritance of strands.

Citation #14 lists by mistake O'Connell among authors. Please check all citations.

Reviewer #2 (Remarks to the Author):

In this work Kishkevich and colleagues have used two previously characterized recombination systems to study what recombination factors are required during the restart of a blocked replication fork and the template switching that is associated with the elongation of the restarted fork. This comparative genetic analysis leads the authors to conclude that recombination-dependent replication (RDR) transitions from an initial event that depends mostly on Rad51 to an elongation phase that relies mostly on Rad52 DNA annealing activity. The work is well done and the general conclusion is supported with the results, although the division of labor is not that strict as suggested. Moreover, this conclusion is not completely novel and restricted mostly to genetic data, lacking molecular evidence to explain how this transition occurs. This main conclusion is accompanied by an extensive genetic characterization of the two reporters that help to visualize shared and specific functions of the initiation and elongation steps of RDR, although it is unclear why some factors are analyzed only in one of the reporters. In conclusion, the study provides an interesting but to some extent confirmatory genetic characterization of the RDR process.

Some points that should be clarified:

- As discussed by the authors, this work is reminiscent of Haber's study showing that the relevance of Rad51 and sequence homology in template switching associated with BIR is reduced as compared to the strand invasion that initiates the process. Thus, the main finding of this study is the requirement of the Rad52 DNA annealing activity; but this result is somehow expected by our knowledge of Rad52, and indeed authors claims in page 15, line 335 "As expected, Rad51-independent RTS1-AO-induced template switching at the 12.4 kb reporter was abolished in a rad51Δ rad52-R45A double mutant indicating its reliance on Rad52's DNA annealing activity". The main observation is that rad52-R45A by itself reduced TS, but this is also expected if Rad51 is not operating.

- The authors claim "that recombination-dependent replication (RDR) is initiated mainly by the Rad51 recombinase, whereas template switching, during the elongation phase of RDR, relies on DNA annealing by Rad52". Although globally the results support a transition in the requirements from the start to the elongation stage, the interpretation of the results is sometimes biased to the conclusion they want to underline. For instance, section 2 and 5 claim that "Ectopic recombination associated with RDR initiation is driven mainly by Rad51" and that "RDR and its associated template switching is only partially dependent on Rad51". The latter argument is based on the lack of effect of rad51Δ on the "12" reporter. But rad51Δ did not affect the frequency of deletions in the "0" reporter either. The major difference is that deletions in the "12" are partially dependent on Rad52 DNA annealing activity, and in "0" both Rad51 and Rad52 DNA annealing activity have to be removed to eliminate deletions. But there is also a synergistic defect in the double mutant rad51Δ rad52-R45A in the frequency of deletions in "12". For some unknown reason the author decide to describe the effect of rad52-R45A in the "12" in a different section, to highlight its relevance in TS, whereas in the "0" they directly compare both mutations in the same section to reinforce the idea that RDR is mostly dependent on Rad51. I suggest the authors to reorganize the description of the results and try to be less categorical by suggesting that RDR is initiated by a mechanism where Rad51 and Rad52 DNA annealing activity have overlapping functions (if RDR initiation were driven mainly by Rad51 its absence should reduce deletions), and transitions to a mechanism where Rad52 DNA annealing activity becomes more relevant. An example of this biased interpretation is in page 18, lines 404-407: "At the site of fork collapse little or no ectopic recombination relies on Rad52's DNA annealing activity, and this is presumably because Rad52's contribution to RDR initiation is adequately compensated for by Rad51". According to the data in Figure 2, the same sentence can be written changing "Rad52's DNA annealing activity" and "Rad51" with each other.

- Some conclusions are drawn from the analyses of mutations in only one of the reporters. This is fine for a characterization of a particular system, but it had been interesting to analyze them in both systems. For instance, what's the role of rad51-3A in the "0" reporter where Rad51 can complement

the absence of Rad52 DNA annealing activity?. Another example; in page 11, lines 241-245. TS is compared with initiation of RDR using data from the *rad51Δ ori-1253Δ* mutant, which is analysed only in the "12" reporter.

- Section 2. "Ectopic recombination associated with RDR initiation is driven mainly by Rad51". The effect of *rad51Δ* in the "0" reporter was already described by the authors, who showed a 3-fold drop in deletions, claiming that Rad51 was required to initiate RDR (Ahn et al EMBO 2005)

- Page 8, l 168. The authors should clarify that this suppressor role of Rad51 is specific of auxiliary mutants, likely as a consequence of a non-functional or incorrectly assembled Rad51/ssDNA nucleofilament (the absence of Rad51 by itself did not increase the frequency of block-induced deletions)

-Pag 12,l 264: *rad51-K300A*; It is *rad54-K300A* (twice)

- Page 21, 462. "As discussed above, *Rad51-3A* is a barrier to Rad51-independent RDR." The effect is mild and, as mentioned in Results, it can be due to its over-expression.

- Discussion: "surprisingly, this inhibition does not correlate with a similar reduction in RTS1-AO-induced template switching at the 12.4 kb reporter. Indeed, unlike at the 0 kb reporter, there is little or no overall reduction in template switch recombination at the 12.4 kb reporter in either a *rad55Δ* or *rdl1Δ* mutant." Authors should be very careful with these comparisons, as "0" and "12" are different reporters. They cannot rule out that RDR initiation in "12" to be Rad55 and Rdl1 independent.

- The paper is a bit long and authors might consider the possibility of reducing some points, in particular speculations from genetic data that might have alternative interpretations and are not important for the main message, mostly related to the description of the auxiliary mutants. Also, the section concerning the Swi5-Sfr1 complex. In *S. cerevisiae* this complex is devoted to meiotic recombination. What is the logic to include it here and why with the "0" reporter only?. The discussion is also too long, with some unnecessary points. Specifically, the section "How does Rad52 initiate RDR and catalyse template switching?" The paper does not provide any data to answer this question.

We thank the reviewers for their time and constructive comments that have helped us to revise and improve our manuscript.

Reviewer #1:

1) The idea that some RDR or BIR events start by the Rad51 pathway but are finished by Rad52-dependent annealing is not novel. Recent work shows that telomere recombination in telomerase deficient cells in budding yeast starts by Rad51 pathway and finishes by mechanism employing Rad52 annealing. This work should be discussed (PMID: 33639094). Despite the similarity of the central conclusion, the work here is important because the recombination type studied is entirely different. It shows that the hybrid mechanism of Rad51-dependent and -independent pathways can likely be applied to different repair contexts.

Our response:

We have included the following text in the Discussion: “A similar transition was recently reported for alternative lengthening of telomeres (ALT) in telomerase defective budding yeast⁶¹. Like RDR, ALT is driven by Rad51-dependent and Rad51-independent pathways that generate so-called type I and type II survivors, respectively. It was thought that these pathways functioned independently of each other however, Malkova and colleagues showed that they instead work together to forge hybrid type I/type II chromosome ends through a unified pathway initiated by Rad51-mediated strand invasion and completed by a Rad51-independent mechanism that relies on Rad52 and Rad59⁶¹.”

2) How does annealing mediate template switch during the elongation phase of RDR? Does the distant ade6-L469 repeat have to be transcribed for the annealing? Would a stronger promoter increase annealing? It is not clear if Alu repeats are transcribed. If not transcription, then how does the second repeat of the reporter become single-stranded? Alternatively, if repair going toward and through the reporters is essential for some supercoiling or unwinding that facilitates annealing, would the repeat located on another chromosome be used for template switching in an annealing-independent manner?

Our response:

The reviewer asks important questions that we aim to investigate in our future research. For the current paper, we have included the following text in the Discussion:

“We have shown that Rad52’s N-terminal DNA binding domain is sufficient to drive RDR and its associated template switching in the absence of Rad51. This domain can promote annealing of complementary ssDNAs and generates D-loops between ssDNA and supercoiled plasmid DNA^{56,66-68}. However, it remains unclear whether its D-loop forming activity works by a Rad51-type strand invasion mechanism, or simply by annealing the ssDNA to patches of ssDNA within the plasmid⁶⁸. In vivo, Rad51-independent BIR is thought to rely on Rad52 annealing the resected end of a DSB to homologous or homeologous ssDNA exposed at a replication fork, transcription complex or secondary DNA structure¹⁵. Rad51-independent RDR in fission yeast may similarly depend on such chance exposed stretches of ssDNA. Alternatively, there may be proteins that work with Rad52 to promote its D-loop forming activity by transiently unwinding DNA. Determining the exact mechanism by which Rad52 initiates RDR and promotes template switching remains an important goal for future research.”

3) The authors propose that during RDR, there is a transition from Rad51-dependent to Rad51-independent pathway. However, they mostly measure template switches close or far from strand invasion. Perhaps template switches occur only at a small fraction of events, and RDR is mostly Rad52 annealing independent. Regular BIR in budding yeast where the synthesis goes for over 100 kb is nearly entirely Rad52-annealing independent (PMID: 33844333). However, template switches were not tested. This work should be discussed.

Our response:

We have added the following text to the Discussion:

“If we assume that the frequency of template switching is an accurate indicator of the amount of RDR, then both pathways appear to hold more equal sway in driving RDR in fission yeast than they do for BIR in budding yeast. Indeed, by comparing the frequency of template switching at the 12.4 kb reporter in wild-type and rad51-3A cells, we can estimate that ~22% of total RDR can be initiated and driven by the Rad51-independent pathway when Rad51 is present and capable of binding DNA. However, we cannot rule out the possibility that template switching is a poor indicator of overall RDR levels, as it might have its own specific genetic requirements and occur in only a minority of RDR events. Therefore, the contribution that the Rad51-independent pathway makes to total RDR remains uncertain. Nevertheless, at least for RDR that is associated with template switching, the Rad51-independent pathway is active and able to make a significant contribution to its initiation and elongation even when Rad51 is present.”

And:

“However, it is important to note that BIR can progress more than 100 kb in budding yeast with seemingly only a small reduction in efficiency when Rad52’s strand annealing activity is absent⁶⁴. Therefore, any transition from Rad51-dependent to Rad51-independent pathway is not essential for the elongation phase of BIR. How important it is for RDR in fission yeast remains to be determined.”

Minor

4) Line 82. BIR is mutagenic mainly because of the inability to use MMR – conservative inheritance of strands.

Our response: We have amended the Introduction as follows:

“This has been attributed to: 1) its conservative mode of DNA synthesis that renders mismatch repair inefficient (Deem et al, 2011; Saini et al, 2013)...”

5) Citation #14 lists by mistake O’Connell among authors. Please check all citations.

We apologise for this mistake caused by a glitch in our EndNote library. We have corrected the citation and checked the others.

Reviewer #2:

Some points that should be clarified:

1) As discussed by the authors, this work is reminiscent of Haber’s study showing that the relevance of Rad51 and sequence homology in template switching associated with BIR is reduced as compared to the strand invasion that initiates the process. Thus, the main finding

of this study is the requirement of the Rad52 DNA annealing activity; but this result is somehow expected by our knowledge of Rad52, and indeed authors claims in page 15, line 335 “As expected, Rad51-independent RTS1-AO-induced template switching at the 12.4 kb reporter was abolished in a *rad51Δ rad52-R45A* double mutant indicating its reliance on Rad52’s DNA annealing activity”. The main observation is that *rad52-R45A* by itself reduced TS, but this is also expected if Rad51 is not operating.

Our response:

Whilst one can make predictions based on published findings, it is nonetheless essential to confirm the validity of these predictions with experimental evidence. Anand et al (2014) showed that there is an increased dependence on Rdh54 for template switching compared to its role in the initiation of BIR. They observed a ~4-fold reduction in ectopic recombination associated with the initiation of BIR and a subsequent ~14-fold reduction in template switching. In comparison, a *rad51Δ* mutant exhibited a ~300-fold reduction in ectopic recombination and a subsequent ~12-fold reduction in template switching. Whilst these data indicate a difference between BIR-associated template switching and BIR initiation, they do not show that these processes transition from being Rad51-dependent to being Rad51-independent. Indeed, the involvement of Rdh54 in promoting template switching does not necessarily imply that template switching is being driven by a Rad51-independent (i.e. Rad52 DNA annealing dependent) pathway. Although Rdh54 functions in Rad51-independent BIR, it also interacts directly with Rad51 and functions alongside Rad54 in governing D-loop dynamics (Crickard, 2021).

Our claim on page 15 (“As expected, Rad51-independent *RTS1*-AO-induced template switching was abolished in a *rad51Δ rad52-R45A* double mutant...”) was only expected after we discovered that template switching is only partially dependent on Rad51. Based on prior studies of BIR in budding yeast, we were expecting to see a much greater dependence on Rad51.

Even after discovering that template switching is only partially dependent on Rad51, one could not predict with certainty that Rad52’s DNA annealing would have such a marked effect on template switching when Rad51 is present. Indeed, there could have been complete redundancy with Rad51.

2) The authors claim “that recombination-dependent replication (RDR) is initiated mainly by the Rad51 recombinase, whereas template switching, during the elongation phase of RDR, relies on DNA annealing by Rad52”. Although globally the results support a transition in the requirements from the start to the elongation stage, the interpretation of the results is sometimes biased to the conclusion they want to underline. For instance, section 2 and 5 claim that “Ectopic recombination associated with RDR initiation is driven mainly by Rad51” and that “RDR and its associated template switching is only partially dependent on Rad51”. The latter argument is based on the lack of effect of *rad51Δ* on the “12” reporter. But *rad51Δ* did not affect the frequency of deletions in the “0” reporter either. The major difference is that deletions in the “12” are partially dependent on Rad52 DNA annealing activity, and in “0” both Rad51 and Rad52 DNA annealing activity have to be removed to eliminate deletions. But there is also a synergistic defect in the double mutant *rad51Δ rad52-R45A* in the frequency of deletions in “12”. For some unknown reason the author decide to describe the effect of *rad52-R45A* in the “12” in a different section, to highlight its relevance in TS, whereas in the “0” they directly compare both mutations in the same section to reinforce the idea that RDR is mostly dependent on Rad51. I suggest the authors to

reorganize the description of the results and try to be less categorical by suggesting that RDR is initiated by a mechanism where Rad51 and Rad52 DNA annealing activity have overlapping functions (if RDR initiation were driven mainly by Rad51 its absence should reduce deletions), and transitions to a mechanism where Rad52 DNA annealing activity becomes more relevant. An example of this biased interpretation is in page 18, lines 404-407: “At the site of fork collapse little or no ectopic recombination relies on Rad52’s DNA annealing activity, and this is presumably because Rad52’s contribution to RDR initiation is adequately compensated for by Rad51”. According to the data in Figure 2, the same sentence can be written changing “Rad52’s DNA annealing activity” and “Rad51” with each other.

Our response:

We have structured our paper in the way we have to highlight the important finding that Rad52’s DNA annealing activity is required for the majority of template switching at the 12.4 kb reporter but is dispensable for ectopic recombination at the 0 kb reporter. This contrasts with loss of Rad51-mediated strand invasion, which has a much bigger effect on the frequency of ectopic recombination at the 0 kb reporter than at the 12.4 kb reporter.

The key observations that support these conclusions are:

- 1) Unlike at the 12.4 kb reporter, the majority (~60%) of ectopic recombination at the 0 kb reporter gives rise to gene conversions. At the 0 kb reporter, the frequency of gene conversions decreases by ~97-fold in a *rad51*Δ mutant, whereas, at the 12.4 kb reporter the reduction is only ~5-fold.
- 2) Whilst it is true that a *rad51*Δ mutant exhibits no statistically significant reduction in deletions at the 0 kb reporter based on our current data, our finding that there is a ~2 – 4-fold reduction in deletions in both *rad55*Δ and *rad57*Δ mutants indicates that Rad51-mediated strand invasion is responsible for the majority of *RTS1*-AO-induced deletions at the 0 kb reporter. This is in stark contrast with the result at the 12.4 kb reporter where there is no reduction in deletions in a *rad55*Δ mutant.
- 3) A *rad52*-R45A mutant exhibits no statistically significant reduction in *RTS1*-AO-induced gene conversions or deletions at the 0 kb reporter, whereas at the 12.4 kb reporter it exhibits a ~2 – 7-fold* reduction in gene conversions and ~5 – 32-fold* reduction in deletions.

(*depending on whether the strain is *ori-1253+* or *ori-1253*Δ)

Based on our data, we think it is reasonable to conclude that Rad51-mediated strand invasion is responsible for the majority of ectopic recombination associated with RDR initiation at the 0 kb reporter and that it plays a lesser role in driving template switching at the 12.4 kb reporter. However, we agree with the reviewer that there is partial redundancy between the Rad51-dependent and Rad51-independent pathways in driving deletions at both the 0 kb and 12.4 kb reporters. What we’re highlighting is which pathway is most likely to be predominant at each site when all factors are present.

We have added the following text to section 2 of the Results to help clarify this issue:

“Altogether our data show that Rad51-mediated strand invasion is the main driver of gene conversions associated with replication fork blockage and RDR initiation. If we assume that fully active Rad51, in the presence of its auxiliary proteins, is just as strong a barrier to Rad51-independent ectopic recombination as Rad51 that is rendered non-functional by mutation or absence of an auxiliary protein, then we can conclude that Rad51-mediated strand invasion is

also the main driver of RTS1-AO-induced deletions. However, even in the presence of non-functional Rad51, some RTS1-AO-induced deletions are formed by a Rad51-independent pathway. This Rad51-independent ectopic recombination, which increases when Rad51 is absent, may be linked to RDR initiation and/or inter-fork strand annealing during fork convergence^{45,46}.”

We have also revised the Discussion to include mention of an alternative model to the “transition hypothesis” in which the Rad51-dependent and -independent pathways function completely separately (Page 18):

“This discovery, that template switching becomes increasingly reliant on Rad52’s DNA annealing activity the further RDR progresses from its initiation site, could be explained by a model in which the Rad51-independent pathway drives RDR elongation at a faster pace than the Rad51-dependent pathway with both pathways remaining completely separate. However, we favour an alternative model in which the catalysis of strand invasion is driven mainly by Rad51 during RDR initiation and then transitions to Rad52 during RDR elongation.”

3) Some conclusions are drawn from the analyses of mutations in only one of the reporters. This is fine for a characterization of a particular system, but it had been interesting to analyze them in both systems. For instance, what’s the role of *rad51-3A* in the “0” reporter where Rad51 can complement the absence of Rad52 DNA annealing activity?. Another example; in page 11, lines 241-245. TS is compared with initiation of RDR using data from the *rad51Δ ori-1253Δ* mutant, which is analysed only in the “12” reporter.

Our response:

Having discovered that mutants from the same protein complex exhibited very similar recombination phenotypes at the 0 kb reporter (e.g. *rad55Δ* and *rad57Δ*, and *rdl1Δ*, *rlp1Δ* and *sws1Δ*), we chose to analyse only one mutant from each of these complexes at the 12.4 kb reporter to make the workload more manageable. These mutants should be representative of the other mutants from the same complex. The *ori-1253Δ* mutant is only used to delay the oncoming replication fork to allow more time for RDR to reach the 12.4 kb reporter and thereby have an opportunity to do template switching. As reported previously, this mutation has a much bigger effect on template switching at the 12.4 kb reporter than at the 0 kb reporter (i.e. there is already sufficient time at the 0 kb reporter to achieve near maximal ectopic recombination) (Nguyen et al, 2015). Therefore, we do not think it is necessary to analyse a *rad51Δ ori-1253Δ* mutant at the 0 kb reporter. However, we do agree with the reviewer that analysis of *rad51-3A* using the 0 kb reporter would be informative and, therefore, we have constructed the necessary strains and performed this experiment. The new data support our conclusion that Rad51 is a partial barrier to Rad51-independent ectopic recombination at the 0 kb reporter and are included on Page 9 of the Results and shown in Supplementary Figure 2 and Supplementary Table 1.

Whilst investigating the effect of the *rad51-3A* mutant on recombination at the 0 kb reporter, we discovered a discrepancy with our data at the 12.4 kb reporter. Specifically, the spontaneous recombination frequencies for *rad51-3A* were different at the two reporters. Whilst the frequency of gene conversions was reduced at both reporters, the frequency of deletions was higher at the 0 kb reporter than at the 12.4 kb reporter. This prompted us to screen further *rad51-3A RTS1-IO* 0 kb and 12.4 kb reporter strain isolates. Originally, we had tested two independent isolates of the *rad51-3A RTS1-IO* 12.4 kb reporter strain both of which exhibited wild-type levels of spontaneous deletions. However, upon screening a

further 12 isolates we found that all exhibited a higher than wild-type level of deletions similar to a *rad51*Δ mutant. Six independent *rad51-3A RTS1-IO* 0 kb reporter strain isolates were also tested and these too exhibited the higher level of deletions similar to a *rad51*Δ mutant. Therefore, we conclude that the true phenotype of the *rad51-3A* mutant for spontaneous recombination is hypo-recombinogenic for gene conversions but hyper-recombinogenic for deletions similar to a *rad51*Δ mutant. We have corrected the data in our revised manuscript and have noted the occurrence of the two atypical isolates in the Reporting Summary document.

4) Section 2. “Ectopic recombination associated with RDR initiation is driven mainly by Rad51”. The effect of *rad51*Δ in the “0” reporter was already described by the authors, who showed a 3-fold drop in deletions, claiming that Rad51 was required to initiate RDR (Ahn et al EMBO 2005)

Our response:

We thank the reviewer for highlighting this discrepancy between our current and previous data. We’re not sure why we’re seeing a higher level of *RTS1-AO*-induced deletions in a *rad51*Δ mutant than we did in our earlier studies. Over the years the source of ingredients for our media has changed and we suspect that this may account for the difference. Importantly, all the strains in our current work have been compared under the same conditions. We have added the following text to Section 2 of the Results to acknowledge the difference between our earlier and current data:

“Whilst the increase in spontaneous deletions and reduction in spontaneous and RTS1-AO induced gene conversions are consistent with published data, previous studies have reported a ~1.5-fold to ~2.5-fold reduction in RTS1-AO-induced deletions in a rad51Δ mutant compared to wild-type³⁷⁻³⁹.”

And:

“Even when compared to previous data³⁷⁻³⁹, the fold reductions in RTS1-AO-induced deletions in rad54Δ, rad54-K300A, rad55Δ, rad57Δ, rdl1Δ, rlp1Δ and sws1Δ mutants are the same or greater than in a rad51Δ mutant. This observation, that a rad51Δ mutant tends to exhibit higher levels of RTS1-AO-induced deletions than its auxiliary factor mutants, suggests that the presence of Rad51 can partially suppress Rad52-mediated recombination as observed in budding yeast¹⁵.”

5) Page 8, l 168. The authors should clarify that this suppressor role of Rad51 is specific of auxiliary mutants, likely as a consequence of a non-functional or incorrectly assembled Rad51/ssDNA nucleofilament (the absence of Rad51 by itself did not increase the frequency of block-induced deletions)

Our response:

We think that Rad51 is likely to suppress Rad51-independent ectopic recombination at the 0 kb reporter both when it’s fully active and when rendered inactive by the loss of one of its auxiliary proteins. We have clarified this point by the addition of a concluding paragraph to Section 2 of the Results (see our response to point 2 above).

6) Page 12,l 264: *rad51-K300A*; It is *rad54-K300A* (twice)

Our response:

Thank you for spotting this mistake – we’ve corrected it.

7) Page 21, 462. “As discussed above, Rad51-3A is a barrier to Rad51-independent RDR.” The effect is mild and, as mentioned in Results, it can be due to its over-expression.

Our response:

We have deleted this sentence.

8) Discussion: “surprisingly, this inhibition does not correlate with a similar reduction in RTS1-AO-induced template switching at the 12.4 kb reporter. Indeed, unlike at the 0 kb reporter, there is little or no overall reduction in template switch recombination at the 12.4 kb reporter in either a rad55 Δ or rdl1 Δ mutant.” Authors should be very careful with these comparisons, as “0” and “12” are different reporters. They cannot rule out that RDR initiation in “12” to be Rad55 and Rdl1 independent.

Our response:

If loss of Rad55 or Rdl1 creates a non-functional Rad51 nucleofilament that is unable to initiate RDR and acts as a partial barrier to Rad51-independent RDR, then we would expect to see a reduction in template switching at the 12.4 kb reporter. The fact that we don’t suggests that either the frequency of template switching for each RDR event increases (which compensates for the reduction in total RDR events) or only ectopic recombination is impaired during RDR initiation. We’ve amended the Discussion to make these points clearer (see page 21 in the revised manuscript).

9) The paper is a bit long and authors might consider the possibility of reducing some points, in particular speculations from genetic data that might have alternative interpretations and are not important for the main message, mostly related to the description of the auxiliary mutants. Also, the section concerning the Swi5-Sfr1 complex. In *S. cerevisiae* this complex is devoted to meiotic recombination. What is the logic to include it here and why with the “0” reporter only?. The discussion is also too long, with some unnecessary points. Specifically, the section “How does Rad52 initiate RDR and catalyse template switching?” The paper does not provide any data to answer this question.

Our response:

We have significantly shortened the description of the 0 kb reporter recombination data for the Rad51 auxiliary factor mutants and combined them into one section (revised section 2, pages 7 – 10). We have retained the data for Swi5-Sfr1 as in fission yeast it has been shown to play a role in mitotic cells. We have also shortened the Discussion section “How does Rad52 initiate RDR and catalyse template switching” (see our response to Reviewer #1’s second point).

REVIEWERS' COMMENTS

Reviewer #1 (Remarks to the Author):

The authors revised the manuscript and addressed most of the questions in satisfactory way by rewriting. The manuscript is now suitable for publication.

Reviewer #2 (Remarks to the Author):

All my concerns have been properly addressed, and the new manuscript is now clearer. As previously mentioned, this is a nice and interesting genetic study with results that support the conclusions and extend the "transition" model from BIR systems to a fork block-associated system (likely more physiological) in a different model organism. This confirmatory feature (beyond the specific characteristics of each system/organism) of the model is not meant to be a handicap for its publication (that is an essential step for science to advance) but somehow affects the novelty of the finding and consequently the journal where it is published (but this is an editorial decision).

Reviewer #1 (Remarks to the Author):

The authors revised the manuscript and addressed most of the questions in satisfactory way by rewriting. The manuscript is now suitable for publication.

Reviewer #2 (Remarks to the Author):

All my concerns have been properly addressed, and the new manuscript is now clearer. As previously mentioned, this is a nice and interesting genetic study with results that support the conclusions and extend the “transition” model from BIR systems to a fork block-associated system (likely more physiological) in a different model organism. This confirmatory feature (beyond the specific characteristics of each system/organism) of the model is not meant to be a handicap for its publication (that is an essential step for science to advance) but somehow affects the novelty of the finding and consequently the journal where it is published (but this is an editorial decision).

Our response:

Neither Reviewer raised any further points for us to address.